# Latent Tuberculosis Infection and Associated Risk Factors among People Living with HIV and HIV-Uninfected Individuals in Lithuania

**DOI:** 10.3390/pathogens12080990

**Published:** 2023-07-28

**Authors:** Elzbieta Matulyte, Zavinta Kancauskiene, Aidas Kausas, Jurgita Urboniene, Vilnele Lipnickiene, Jelena Kopeykiniene, Tomas Gudaitis, Sarunas Raudonis, Edvardas Danila, Dominique Costagliola, Raimonda Matulionyte

**Affiliations:** 1Department of Infectious Diseases and Dermatovenerology, Faculty of Medicine, Vilnius University Hospital Santaros Klinikos, Vilnius University, LT-08410 Vilnius, Lithuania; raimonda.matulionyte@santa.lt; 2Department of Infectious Diseases, University Hospital of Klaipeda, LT-92888 Klaipeda, Lithuania; zavinta.kancauskiene@yahoo.com; 3Adult Infectious Diseases Unit, Clinic of Conservative Medicine, Republican Siauliai County Hospital, LT-76231 Siauliai, Lithuania; aidas.kausas@siauliuligonine.lt; 4Centre of Infectious Diseases, Vilnius University Hospital Santaros Klinikos, LT-08410 Vilnius, Lithuania; jurgita.urboniene@santa.lt; 5National Public Health Surveillance Laboratory, LT-10210 Vilnius, Lithuania; vilnele.lipnickiene@nvspl.lt; 6Department of Diagnostics, University Hospital of Klaipeda, LT-92888 Klaipeda, Lithuania; j.kopeykiniene@kul.lt; 7Faculty of Medicine, Vilnius University, LT-03101 Vilnius, Lithuania; tomas.gudaitis@mf.stud.vu.lt (T.G.); sarunas.raudonis@mf.stud.vu.lt (S.R.); 8Faculty of Medicine, Institute of Clinical Medicine, Clinic of Chest Diseases, Immunology, and Allergology, Vilnius University, LT-03101 Vilnius, Lithuania; edvardas.danila@santa.lt; 9Centre of Pulmonology and Allergology, Vilnius University Hospital Santaros Klinikos, LT-08661 Vilnius, Lithuania; 10Institut Pierre Louis Épidémiologie de Santé Publique, Sorbonne Université, INSERM, F75013 Paris, France; dominique.costagliola@iplesp.upmc.fr

**Keywords:** LTBI, HIV, risk factors, prevalence, TB

## Abstract

Background: People living with HIV (PLHIV) with latent tuberculosis infection (LTBI) are at increased risk of tuberculosis (TB) reactivation compared to the HIV-negative population. Lithuania belongs to the 18 high-priority TB countries in the European region. The aim of this study was to compare the prevalence of LTBI and LTBI-related risk factors between PLHIV and HIV-uninfected populations. Methods: A cross-sectional study was conducted in three Lithuanian Infectious Diseases centres from August 2018 to May 2022 using the interferon gamma release assay (IGRA) and tuberculin skin test (TST) in Vilnius, and IGRA only in Siauliai and Klaipeda. Cohen’s kappa was used to assess IGRA and TST agreement. A structured questionnaire was completed by the study participants. LTBI-related risk factors were identified using a multivariable logistic regression model. Results: In total, 391 PLHIV and 443 HIV-uninfected individuals enrolled, with a median age of 41 (IQR 36–48) and 43 (IQR 36–50), consisting of 69.8% and 65.5% male, respectively. The prevalence of LTBI defined by positive IGRA and/or TST among PLHIV was higher compared to that in the HIV-uninfected population (20.5% vs. 15.3%; OR 1.42; 95% CI 1.02–2.03; *p* = 0.04). The concordance between IGRA and TST was fair: kappa = 0.23 (95% CI 0.09–0.34). In multivariable analyses, association with injecting drug use (IDU) (ORa 2.25, 95% CI 1.27–3.99, *p* = 0.01) and imprisonment (ORa 1.99, 95% CI 1.13–3.52, *p* = 0.02) in all participants, IDU (ORa 2.37, 95% CI 1.09–5.15; *p* = 0.029) in PLHIV and a history of contact with an active TB patient (ORa 3.33, 95% CI 1.53–7.24; *p* = 0.002) in HIV-uninfected individuals were significant associations evidenced by LTBI. Conclusions: The prevalence of LTBI among PLHIV in Lithuania is higher compared to that in the HIV-uninfected population and the European average. The association with IDU in PLHIV emphasizes the need for integrated HIV, TB and substance abuse treatment to provide patient-centred care.

## 1. Introduction

The World Health Organization (WHO) estimated that 8% of the 9.9 million people worldwide who developed tuberculosis (TB) in 2020 were those living with human immunodeficiency virus (HIV) [1]. Additionally, the WHO highlighted that, for the first time in over a decade, the TB death rate has increased because of decreased access to prevention and care due to the COVID-19 pandemic. Mortality related to acquired immune deficiency syndrome (AIDS) remains high in Eastern Europe, and pulmonary TB is the most common AIDS-defining condition in this region [2]. A considerable proportion of people living with HIV (PLHIV) in Eastern Europe have a delayed diagnosis of TB, leading to a higher risk of death, and TB remains undiagnosed at death for a number of PLHIV [3].

Latent tuberculosis infection (LTBI) is defined as a state of persistent immune response to stimulation by *Mycobacterium tuberculosis* antigens with no evidence of clinically manifested active TB [4]. WHO estimated that 1.8 billion people, or one-third of the world’s population, had LTBI in 1999 [5]. In 2016, the global prevalence of LTBI was updated to 23%, which corresponds to 1.7 billion people infected worldwide [6]. Studies have shown that <3% of these people will develop active TB over the course of their lifetime, the risk being 20–100 times higher among PLHIV [4,7,8,9,10,11].

TB continues to be a public health problem in Lithuania, as one of the 18 high-priority TB countries in the European region. Lithuania has one of the highest rates of TB in the European region (ranging from 40.7/10,000 in 2018 to 26.3/100,000 in 2022) [1,12,13,14,15,16] and out of the cumulative number of all 3431 PLHIV in Lithuania at the end of 2020 [17], 311 (9.1%) patients were registered with active TB [18]. TB is the most common AIDS-defining condition in Lithuania; its proportion has been the highest in the EU/EEA since 2015 and was still >50% of all AIDS-defining conditions in 2019 [19].

The prevention of active TB disease by the treatment of LTBI is a critical component of the WHO End TB Strategy. Until June 2018, PLHIV were not screened for LTBI in Lithuania. Therefore, the prevalence is neither known among PLHIV nor in the general population, and no special actions have been taken to prevent active illness. Therefore, the aim of our study was to describe and compare the prevalence of LTBI and LTBI-related risk factors between PLHIV and the HIV-uninfected population.

## 2. Materials and Methods

### 2.1. Study Design, Setting and Subjects

A cross-sectional study was conducted from August 2018 to May 2022 in three geographically distinct infectious disease departments/centres: Vilnius University Hospital Santaros Klinikos, Republican Siauliai County Hospital and Klaipeda University Hospital. These hospitals provide the vast majority of primary, outpatient and inpatient care to PLHIV in Lithuania. The provision of care for PLHIV is funded by the mandatory health insurance fund budget and all HIV centres, including study centres, are public in Lithuania. The screening for LTBI using IGRA among PLHIV has been carried out in Lithuania since June 2018, and it includes the screening of newly diagnosed PLHIV. Screening for active TB among PLHIV is available and is offered to a patient if there are clinical signs of active TB. All patients with active TB are offered an HIV test.

PLHIV 18–65 years of age, who made their first visit after 1 January 2012 to one of the HIV centres participating in the study, were enrolled consecutively as they presented for a scheduled visit; 128/390 (32.8%) were newly diagnosed with HIV. The comparison group, matched to PLHIV by age and gender, was made up of HIV-uninfected participants 18–65 years of age that met the following criteria:(1)Hospitalized or consulting for any reason except HIV in Vilnius University Hospital Infectious Diseases Centre;(2)Consulting for any reason except HIV in the outpatient clinic of the Republican Siauliai County Hospital Infectious Diseases Department;(3)Consulting for any reason except HIV in the outpatient clinic of Klaipeda University Hospital Infectious Diseases Department.

The five most common illness groups, as classified by the International Statistical Classification of Diseases and Related Health Problems 10th Revision, Australian Modification (ICD-10-AM), for which enrolled HIV-uninfected patients were consulting were as follows:-Screening for infectious and parasitic diseases (Z11), 50.8% of patients;-Viral hepatitis (B15–B19), 20.8% of patients;-Viral infections of the central nervous system (A80–A89), 7.5% of patients;-Lyme disease (A69.2), 3.5% of patients;-Fever of unknown origin (R50.9), 3% of patients.

PLHIV and HIV-uninfected individuals were excluded if they met any of the following criteria:-With active TB or past history of TB;-Pregnant or breastfeeding women;-Persons who could not give informed consent owing to a mental disorder.

### 2.2. Ethical Statement

The study was approved by the Regional Biomedical Research Ethics Committee of Vilnius University (3 December 2019, No. 2019/12-1167-658). Informed consent was obtained from all participants. The information obtained was made anonymous and de-identified prior to analysis to ensure confidentiality.

### 2.3. Data Collection and Assessment of LTBI

A structured data collection questionnaire was prepared for the interview of participants to gather information related to sociodemographic characteristics and to extract the data from patients’ medical records. It contained demographic data, smoking status, alcohol consumption, injecting drug use (IDU), history of incarceration, history of Bacillus Calmette et Guerin (BCG) vaccination defined by the presence of a BCG scar, close contact with an active TB patient (household or non-household), timing of HIV diagnosis, CD4 cell count and HIV viral load at time of HIV diagnosis and at time of enrolment, co-infection with hepatitis C virus (HCV) defined by positive HCV IgG antibody (anti-HCV), hepatitis B virus (HBV) defined by positive hepatitis B surface antigen (HBsAg) and ART initiation date. Close contact with an active TB patient was defined according to WHO recommendations for investigating contacts of persons with infectious TB: (1) a household contact—a person who shared the same living space ≥3 months before enrolment with an active TB patient; or (2) a non-household contact—a person who shared a living or working space with an active TB patient [20]. Until February 2018, the HIV detection limit in Lithuania was 75 copies/mL, and, after February, 2018 it was 40 copies/mL. An HIV-RNA viral load of <75 copies/mL or <40 copies/mL was counted as 74 copies/mL and 39 copies/mL, respectively.

Participants were tested for LTBI using interferon gamma release assay (IGRA) followed by tuberculin skin test (TST) in Vilnius and with IGRA only in Klaipeda and Siauliai. QuantiFERON-TB Gold In-Tube test (QFT-GIT; Qiagen, Hilden, Germany) was used in Vilnius and Siauliai centres, and LIOFeron TB/LTBI (Lionex, Braunschweig, Germany) in Klaipeda centre. The results were classified as recommended by the manufacturers for IGRA. QFT-GIT and LIOFeron TB/LTBI were interpreted as positive if the IFN-γ was ≥0.35 IU/mL and ≥0.2 IU/mL, respectively. The TST was performed by an intradermal injection with 5 tuberculin units (TU) of tuberculin PPD (BB-NCIPD Ltd., Sofia, Bulgaria). The skin test reaction after TST was evaluated 48–72 h after tuberculin administration: induration ≥5 mm was defined as positive.

### 2.4. Primary and Secondary Outcomes

The primary study outcome was LTBI prevalence defined by a positive IGRA and/or TST result. Secondary outcomes were in agreement between IGRA and TST, as were risk factors for LTBI.

### 2.5. Statistical Analysis

Frequencies, proportions and summary statistics were used to describe the study population in relation to sociodemographic and clinical characteristics. Categorical variables were analysed using Pearson chi-square and Fisher exact tests when appropriate. Continuous variables were expressed as the median and interquartile range (IQR). The nonparametric Mann–Whitney U test was used to identify differences between groups in continuous outcomes. Cohen’s kappa (kappa) was used to assess IGRA and TST agreement. The strength of agreement was considered slight if kappa ≤ 0.20, fair if kappa = 0.21–0.40, moderate if kappa = 0.41–0.60, substantial if kappa = 0.61–0.80 and optimal if kappa = 0.81–1.00 [21]. Univariable logistic analysis was used to explore the unadjusted association between variables (sociodemographic characteristics, imprisonment, smoking, any-time intravenous drug use, alcohol abuse, history of contact with a TB patient, BCG vaccination status, hepatitis status, HIV status, CD4 count and HIV viral load at the time of HIV diagnosis and IGRA test) and outcome. Only variables with a *p*-value < 0.10 in any of the two univariable analyses of PLHIV and the HIIV-uninfected population were included in the multivariable analysis, except for hepatitis C because of its close association with intravenous drug use, and CD4 count, available only for PLHIV. Analysis was conducted using IBM SPSS version 20.0.

## 3. Results

### 3.1. Population Characteristics

From August 2018 to May 2022, 834 participants were enrolled: 391 PLHIV and 443 HIV-uninfected individuals. Among PLHIV, all patients were Caucasian, 272 (69.8%) were male and the median age was 41 (IQR 36–48). Among the HIV-uninfected individuals, all patients were Caucasian, 290 (65.5%) were male and the median age was 43 (IQR 36–50). The sociodemographic and clinical characteristics of study participants are shown in Table 1.

Among PLHIV, 79.0% had no university education, 38.1% were unemployed and 36.3% had a history of imprisonment. In total, 14% had had contact with an active TB patient and 90.0% had a previous BCG vaccination. Smoking, IDU and abuse of alcohol were identified in 62.4%, 39.4% and 7.9% of participants, respectively. Hepatitis C was found in 46.6% of PLHIV; 98.1% of them had a history of IDU.

Among the HIV-uninfected individuals, 67.3% had no university education, 13.8% were unemployed and 9.7% had a history of imprisonment. In total, 7.9% had had contact with an active TB patient and 95.0% had a previous BCG vaccination. Smoking, IDU and abuse of alcohol were identified in 33.6%, 8.6% and 4.1% of participants, respectively. Hepatitis C was found in 46.5% of HIV-uninfected patients; 94.7% of them had a history of IDU.

Regarding HIV-related characteristics, the median CD4 count at the time of HIV diagnosis was 317 (IQR 170–508) cells/mm^3^ and the median HIV-RNA was 21,250 (IQR 2978–107,000) copies/mL. At the time of the IGRA test, the median CD4 count was 475 (IQR 269–673) cells/mm^3^, the median HIV-RNA was 39 (IQR 0–9893) copies/mL and 60.4% of patients had a viral load lower than 200 copies/mL. Overall, 67.2% of patients were on ART at the time of the IGRA test, and the median time of ART was 35.2 (IQR 0.3–62.8) months.

Median CD4 count at the time of the IGRA/TST test was lower in the Infectious Diseases Department of Klaipeda compared to Vilnius (343 (IQR 182–561) cells/mm^3^ vs. 496 (IQR 287–674) cells/mm^3^, *p* < 0.001) and compared to Siauliai (343 (IQR 182–561) cells/mm^3^ vs. 617 (386–799) cells/mm^3^, *p* < 0.001) (Table 2).

### 3.2. Prevalence of LTBI

The prevalence of LTBI defined by positive IGRA and/or TST among PLHIV was higher compared to the HIV-uninfected population: 20.5% vs. 15.3%; OR 1.42; 95% CI 1.02–2.03; *p* = 0.04. The QFT-GIT test was carried out for 649 participants in Vilnius and Siauliai: 316 in PLHIV and 333 in HIV-negative individuals. The positive QFT-GIT was found in 64 (20.3%) PLHIV. In the group of HIV-uninfected individuals, positive QFT-GIT was found in 31 (9.3%) cases. LIOFeron TB/LTBI was carried out for 184 participants in Klaipeda: 74 PLHIV and 110 HIV-negative persons. The positive LIOFeron TB/LTBI was found in 6/74 (8.1%) cases of PLHIV and in 18/110 (12.8%) HIV-negative persons. Among PLHIV, the rate of positive IGRA was higher in Siauliai (25.4%) compared to Vilnius (19.1%) and Klaipeda (8.1%) (*p* = 0.03) (Table 2). TST was carried out for participants in the Vilnius centre: 254 PLHIV and 201 HIV-uninfected participants. The rate of positive TST was 22/243 (9.1%) in PLHIV and 24/198 (12.1%) in HIV-uninfected individuals (*p* = 0.30).

### 3.3. Factors Associated with LTBI in All Study Participants

The association between LTBI and the sociodemographic and clinical characteristics of the overall population was assessed through univariable and multivariable analyses.

In univariable analysis, non-university educational status (OR 1.66, 95% CI 1.07–2.57, *p* = 0.02), unemployment (OR 2.02, 95% CI 1.39–2.95, *p* < 0.001), a history of contact with an active TB patient (OR 1.82, 95% CI 1.10–3.01, *p* = 0.02), imprisonment (OR 3.55, 95% CI 2.43–5.19, *p* < 0.001), hepatitis C co-infection (OR 2.92, 95% CI 1.88–4.52, *p* < 0.001), smoking (OR 1.89, 95% CI 1.32–2.72, *p* = 0.001), IDU (OR 3.57, 95% CI 2.45–5.21, *p* < 0.001) and homelessness (OR 4.09, 95% CI 1.36–12.36, *p* = 0.01) were associated with LTBI.

Multivariable analysis revealed that IDU (ORa 2.25, 95% CI 1.27–3.99, *p* = 0.01) and imprisonment (ORa 1.99, 95% CI 1.13–3.52, *p* = 0.02) were risk factors for LTBI (Figure 1).

### 3.4. Factors Associated with LTBI among PLHIV

Both univariable (Table 1) and multivariable analyses (Figure 1) were carried out to identify the association between LTBI and the sociodemographic and clinical characteristics of PLHIV.

In the univariable analysis, non-university educational status (OR 3.99, 95% CI 3.99 1.67–9.532, *p* = 0.002), IDU (OR 4.71, 95% CI 2.77–8; *p* < 0.001), hepatitis C co-infection (OR 6.01, 95% CI 3.22–11.23; *p* < 0.001), imprisonment (OR 4.25, 95% CI 2.53–7.13; *p* < 0.001), smoking (OR 3.2, 95% CI 1.75–5.85; *p* < 0.001), unemployment (OR 2.73, 95% CI 1.65–4.51; *p* < 0.001) and CD4 count >350 cells/mm^3^ at time of HIV diagnosis (OR 22.44, 95% CI 1.43–4.14; *p* = 0.001) were associated with LTBI (Table 1). IDU (ORa 2.37, 95% CI 1.09–5.15; *p* = 0.029) was the only significant association with LTBI found in the multivariable analysis (Figure 1).

### 3.5. Factors Associated with LTBI among HIV-Uninfected Individuals

The association between LTBI and the sociodemographic and clinical characteristics of HIV-uninfected individuals was assessed through univariable (Table 1) and multivariable analysis (Figure 1).

In univariable analysis, imprisonment (OR 2.72, 95% CI 1.34–5.53; *p* = 0.006), IDU (OR 2.49, 95% CI 1.17–5.29; *p* = 0.018) and a history of contact with an active TB patient (OR 3.28, 95% CI 1.54–6.96; *p* = 0.002) were associated with LTBI (Table 1). In multivariable analysis, a history of contact with an *M. tuberculosis*-positive TB patient (ORa 3.33, 95% CI 1.53–7.24; *p* = 0.002) was the only significant association with LTBI found (Figure 1).

### 3.6. Agreement between IGRA and TST

Excluding those with missing TST results, 441 had valid TST and QFT-GIT results. Of these, 350 (79.4%) were in agreement with negative TST and QFT-GIT results and 17 (3.8%) were in agreement with positive TST and QFT-GIT results (Figure 2). The overall agreement (positive and negative) of QFT-GIT and TST was fair: kappa = 0.23 (95%CI 0.09–0.34) (Table 3).

## 4. Discussion

This study compared LTBI prevalence in PLHIV and HIV-uninfected adults and identified risk factors associated with LTBI in a country with a low incidence of HIV, an intermediate prevalence of TB, and a high BCG vaccination coverage.

We found that the prevalence of LTBI, defined by a positive IGRA test and/or TST, was higher among PLHIV (20.5%) than in HIV-uninfected adults (15.3%). The prevalence of LTBI among PLHIV in this study was higher compared to the WHO European region estimates for 1999 (15%) and to the new estimates reported in 2019 by Cohen A. et al. for the WHO European region, defined as 12.2% [5,22]. On the other hand, previous studies reported that PLHIV were less likely to have a positive IGRA result than the HIV-uninfected population [23,24]. These differences could be explained by the low CD4 counts: the results of IGRA depend predominantly on the CD4 recognition of *M. tuberculosis* antigens [25,26,27,28,29]. In our study, 65.5% of PLHIV had a CD4 count ≥350 cells/mm^3^ and had higher IGRA-positive rates compared to those with a CD4 count <350 cells/mm^3^, but these findings were significant by univariable analysis only. Furthermore, more than half of PLHIV were on ART at the time of the IGRA test with an immune status improvement and a reduced probability of false-negative IGRA results as shown in other studies [30,31,32,33]. Therefore, the higher prevalence of LTBI among PLHIV emphasizes the need for the treatment of LTBI in this population and the consideration of the possibility to repeat IGRA in patients with an initially low CD4 count and a negative IGRA result.

A different prevalence of LTBI defined by a positive IGRA test among PLHIV was observed in the study centres: it was higher in Siauliai and Vilnius (25.4% and 19.1%, respectively) compared to Klaipeda (8.1%). The different IGRA positivity rates may be due to the variations in the reported incidence of active TB in the related counties: higher trends of active TB incidence were observed in Siauliai compared to Vilnius and Klaipeda [16,16]. However, an unproportionally lower prevalence of LTBI in Klaipeda might be associated with a lower median CD4 count at the time of the IGRA test compared to Siauliai and Vilnius, while the difference was not found in HIV-uninfected individuals.

Within our study population, PLHIV were less likely to have a positive TST than HIV-uninfected persons, even though a significant association has not been found. The findings echoed the results of studies where HIV-uninfected individuals had higher rates of positive TST results than PLHIV [23,34,35,36,37]. These differences can be explained by the skin test anergy to tuberculin, which is a well-documented phenomenon among PLHIV and is associated with low CD4 cell counts [35]. The majority of study participants (90% of PLHIV and 95% of HIV-uninfected individuals) had a history of BCG vaccination. The BCG vaccination is known to cause false-positive TST results from cross-reactions with mycobacterial antigens [36]. In a retrospective cohort study involving over 3000 participants in the USA assessing the effect of the BCG vaccine on TST reactivity, a positive TST result was associated with BCG vaccination, and it was found that the effect of BCG vaccination on TST reactivity could extend up to 55 years after vaccination [37]. In our study, an induration of at least 5 mm was considered a positive TST. Several studies have demonstrated that increasing the TST cut-off value from 5 mm to 10 mm for HIV-uninfected persons resulted in better specificity with an undefined impact on sensitivity [38,39]. These results highlight that the immune status, history of BCG vaccination and cut-off values should be taken into account when interpreting TST results.

In our study, the discordant results of QFT-GIT and TST led to only a fair agreement between the two tests among PLHIV and a slight agreement in the HIV-negative population. The discrepancies between TST and IGRA test results in PLHIV and HIV-uninfected individuals may be explained as false-negative TST results among PLHIV due to immunosuppression and false-positive results in HIV-negative persons due to a prior BCG vaccination. The findings are largely in agreement with other published studies conducted in intermediate/high-burden TB regions with high coverage of BCG vaccination, comparing populations of PLHIV and HIV-uninfected individuals [38,39,40,41,42]. The feasibility of TST was also limited by operational disadvantages such as tuberculin storage requirements and a need for a patient recall, and a partial (96.9%) evaluation of the results in our study. Therefore, our study findings suggest that IGRA rather than TST should be considered for the testing of LTBI in the Lithuanian PLHIV population.

Our study showed that IDU and imprisonment were associated with LTBI in the whole study population. In separate groups, PLHIV and HIV-uninfected individuals, we found that IDU is a significant risk factor for LTBI among PLHIV, and a history of contact with an active TB patient is a risk factor among HIV-uninfected individuals. Close contact with an active TB patient has been shown to be a risk factor in previous studies [43,44,45,46,47]. Imprisonment is commonly the primary risk factor for TB or an additional factor for TB evolution in the population with other risk factors, i.e., substance abuse and HIV infection. IDU is a known risk factor for LTBI due to the increased possibility of exposure in a crowded environment and incarceration [23,28,48]. In our study, IDU was associated with LTBI in the whole study population and among PLHIV only, and by univariable analysis was partially associated with LTBI even in the HIV-negative population. The majority of study participants with a history of IDU were HCV-infected: 98.1% of PLHIV and 94.7% of HIV-uninfected individuals. The association of IDU and LTBI among PLHIV and the high rates of co-infection with HCV among intravenous drug users support the statement that IDU is a common driver of the HIV, TB and HCV *syndemic* in Eastern Europe [49]. IDU and the importance of harm reduction programmes are broadly described among PLHIV, while LTBI and TB-related healthcare requirements are not documented widely. Therefore, our study findings emphasize the importance of integrated HIV, TB, HCV and substance abuse treatment services, including evaluation for LTBI, to provide person-centred rather than disease-centred care.

The strength of this study is that it provides a relatively precise snapshot of the prevalence of LTBI among PLHIV in Lithuania, as it was a cross-sectional study conducted in three geographically distinct Infectious Diseases Departments in Lithuania. Additionally, the results of this study enable us to identify country-specific circumstances that should be considered for the implementation of the programmatic management of LTBI in PLHIV in Lithuania. We hope that this study will have additional value in reducing barriers for PLHIV with regard to testing for LTBI and raising awareness about LTBI among the patient population and health care providers.

Our study has several limitations. The presence of LTBI was established by two different molecular methods in Vilnius University Hospital and Republican Siauliai County Hospital (QuantiFERON-TB Gold In-Tube) and Klaipeda University Hospital (LIOFeron TB/LTBI). This raises the question of a difference in sensitivity of the two tests, also accounting for different levels of immunodepression in the participants enrolled in the three centres; nevertheless, both tests are validated and used in daily practice. Furthermore, the sample size in our study was relatively small, which may have limited the power to assess the factors associated with LTBI. Finally, TST and IGRA were performed only once for each patient, hindering the possibility of evaluating future test conversion, especially in PLHIV, particularly among those who had discordant results between tests.

## 5. Conclusions

The prevalence of LTBI among PLHIV in Lithuania is higher compared to that in the HIV-uninfected population and the European average. The association found with IDU in PLHIV defines a key population and emphasizes the need for integrated HIV, TB and substance abuse treatment, starting with evaluation for LTBI, with the aim to ensure patient-centred care.

## Figures and Tables

**Figure 1 pathogens-12-00990-f001:**
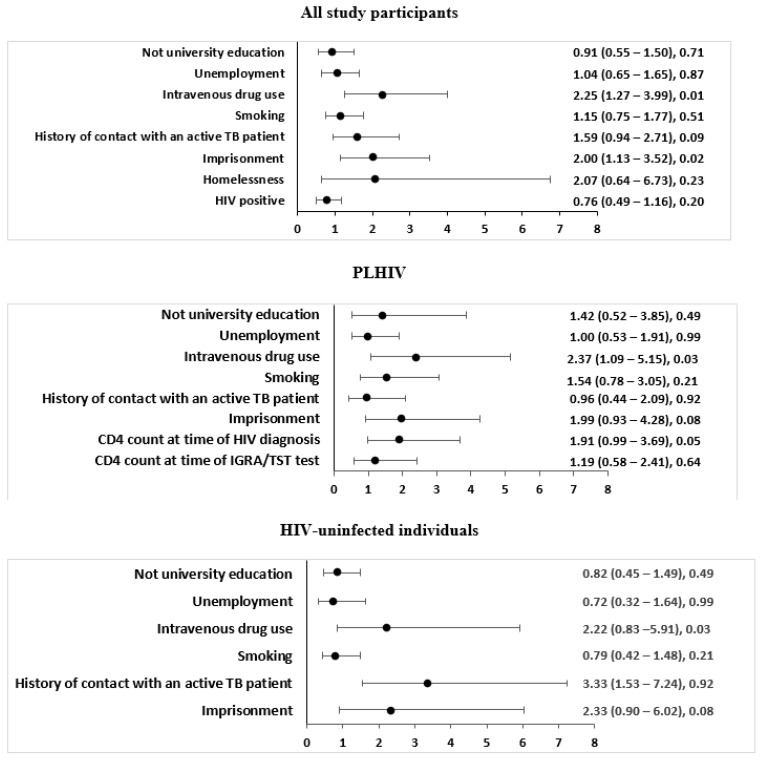
Multivariable analysis of factors associated with LTBI among all study participants, PLHIV and HIV-uninfected individuals in Lithuania. LTBI: latent tuberculosis infection; PLHIV: people living with human immunodeficiency virus; CI: confidence interval; CD4: cluster differentiation-4; HIV: human immunodeficiency virus; TB: tuberculosis. Only variables with a *p*-value < 0.10 in either of the two univariable analyses of PLHIV and HIV-uninfected individuals were included in the multivariable analysis, except for hepatitis C because of its close association with intravenous drug use, and CD4 count, available only for PLHIV. The study population model includes 834 patients. The PLHIV model includes 363 patients, with 28 patients excluded due to missing data. The HIV-uninfected individual model includes 443 patients.

**Figure 2 pathogens-12-00990-f002:**
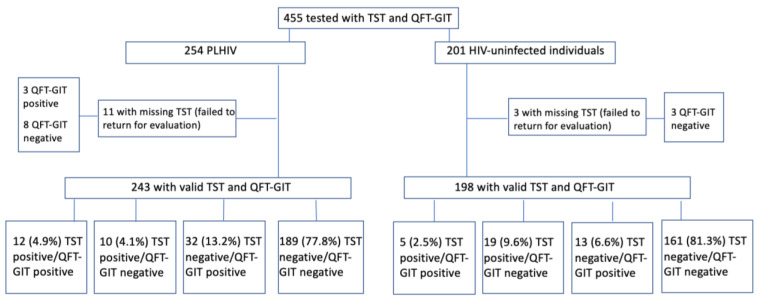
Flow chart showing the numbers of patients tested with TST and QFT-GIT in Vilnius University Hospital. PLHIV: people living with human immunodeficiency virus; HIV: human immunodeficiency virus; QFT-GIT: Quantiferon-TB Gold in-Tube; TST: tuberculin skin test.

**Table 1 pathogens-12-00990-t001:** Sociodemographic and clinical features of the enrolled patients according to LTBI status in PLHIV and HIV-uninfected individuals (*n* = 834).

Characteristic		PLHIV (*n* = 391)				HIV-Uninfected Individuals (*n* = 443)			
		LTBI (*n* = 80), *n* (%)	No LTBI (*n* = 311), *n* (%)	Crude OR (95%CI)	*p*-Value	LTBI (*n* = 68), *n* (%)	No LTBI (*n* = 375), *n* (%)	Crude OR (95%CI)	*p*-Value
Gender	Male	58 (72.5)	215 (69.1)	1.18 (0.68–2.03)	0.559	46 (67.7)	244 (65.1)	1.12 (0.65–1.95)	0.681
	Female	22 (27.5)	96 (30.9)			22 (32.3)	131 (34.9)		
Age	≤40 years	43 (53.8)	147 (47.3)	1.30 (0.79–2.12)	0.301	31 (45.6)	151 (40.3)	1.24 (0.74–2.09)	0.412
	>40 years	37 (46.2)	164 (52.7)			37 (54.4)	224 (59.7)		
Educational status	Not University	74 (92.5)	235 (75.6)	3.99 (1.67–9.53)	0.002	45 (66.2)	253 (67.5)	0.94 (0.55–1.63)	0.835
	University	6 (7.5)	76 (24.4)			23 (33.8)	122 (32.5)		
Unemployment	Yes	46 (57.5)	103 (33.1)	2.73 (1.65–4.51)	<0.001	9 (13.2)	52 (13.9)	0.95 (0.44–2.03)	0.889
	No	34 (42.5)	208 (66.9)			59 (86.8)	323 (86.1)		
History of contact with an active TB patient	Yes	12 (15)	43 (13.8)	1.10 (0.55–2.20)	0.788	12 (17.6)	23 (6.1)	3.28 (1.54–6.96)	0.002
	No/unknown	68 (85)	268 (86.2)			56 (82.4)	352 (93.9)		
Imprisonment history	Yes	51 (63.8)	91 (29.3)	4.25 (2.53–7.13)	<0.001	13 (19.1)	30 (8)	2.72 (1.34–5.53)	0.006
	No	29 (36.2)	220 (70.7)			55 (80.9)	345 (92)		
BCG-vaccinated	Yes	73 (91.3)	279 (89.7)	1.20 (0.51–2.82)	0.682	64 (94.1)	356 (95.2)	0.81 (0.27–2.47)	0.710
	No	7 (8.7)	32 (10.3)			4 (5.9)	18 (4.8)		
Anti-HCV positive (*n* = 556)	Yes	55 (79.7)	96 (37.6)	6.01 (3.22–11.23)	<0.001	22 (50)	86 (45.7)	1.19 (0.61–2.29)	0.611
	No	14 (20.3)	159 (62.4)			22 (50)	102 (54.3)		
HBsAg positive (*n* = 487)	Yes	3 (4.6)	9 (3.8)	1.22 (0.32–4.62)	0.775	2 (7.1)	27 (16.9)	0.38 (0.08–1.68)	0.200
	No	62 (95.4)	226 (96.2)			26 (92.9)	132 (83.1)		
Smoking	Yes	65 (81.2)	179 (57.6)	3.20 (1.75–5.85)	<0.001	24 (35.3)	125 (33.3)	1.09 (0.63–1.88)	0.753
	No	15 (18.8)	132 (42.4)			44 (64.7)	250 (66.7)		
Intravenous drug use	Ever	55 (68.8)	100 (32.2)	4.71 (2.77–8.00)	<0.001	11 (16.2)	27 (7.2)	2.49 (1.17–5.29)	0.018
	Never	25 (31.2)	212 (67.8)			57 (83.8)			
Alcohol abuse	Yes	6 (7.5)	25 (8)	0.93 (0.37–2.34)	0.874	3 (6.1)	15 (3.8)	1.61 (0.51–5.05)	0.413
	No	74 (92.5)	286 (92)			46 (93.9)	379 (96.2)		
BMI, kg/m^2^	<18.5	2 (2.5)	11 (3.5)	0.70 (0.15–3.22)	0.646	1 (2.0)	7 (1.8)	1.86 (0.37–9.43)	0.452
	≥18.5	78 (97.5)	300 (96.5)			48 (98.0)	387 (98.2)		
CD4 count at time of HIV diagnosis, cells/mm^3^ (*n* = 366)	≤350	28 (39.4)	181 (61.4)			-	-	-	-
	>350	43 (60.6)	114 (38.6)	2.44 (1.43–4.14)	0.001				
CD4 count at time of IGRA/TST test, cells/mm^3^ (*n* = 385)	≤350	20 (25)	114 (37.4)			-	-	-	-
	>350	60 (75)	191 (62.6)	1.79 (1.03–3.12)	0.040				
HIV RNA at time of HIV diagnosis, copies/mL (*n* = 338)	<200	8 (12.1)	24 (8.8)			-	-	-	-
	≥200	58 (87.9)	248 (91.2)	0.70 (0.30–1.64)	0.414				
HIV RNA at time of IGRA/TST test, copies/mL (*n* = 380)	<200	46 (57.5)	183 (61)						
	≥200	34 (42.5)	117 (39)	1.16 (0.70–1.91)	0.570	-	-	-	-
On ART (*n* = 381)	Yes	49 (61.3)	201 (66.8)	0.79 (0.47–1.31)	0.356	-	-	-	-
	No	31 (38.7)	100 (33.2)						

PLHIV: people living with human immunodeficiency virus; HIV: human immunodeficiency virus; LTBI: latent tuberculosis infection; OR: odds ratio; CI: confidence interval; TB: tuberculosis; BCG: Bacillus Calmette et Guerin; HCV: hepatitis C virus; HBsAg: hepatitis B surface antigen; CD4: cluster differentiation-4; IGRA: interferon gamma release assay; RNA: ribonucleic acid; ART: antiretroviral therapy.

**Table 2 pathogens-12-00990-t002:** Prevalence of LTBI and related characteristics in the study centres.

Centre	Vilnius	Siauliai	Klaipeda	*p*-Value
TB incidence per 100,000 population in the county in 2019–2021, ranges [16]	17.7–20.6	33.5–37.7	23.4–38.1	-
Enrolled PLHIV	*n* = 258	*n* = 59	*n* = 74	-
- Median CD4 at the time of IGRA/TST, cells/mm^3^ (IQR)	495 (287–674)	617 (386–799)	343 (182–561)	<0.001 *
Enrolled HIV-uninfected individuals	*n* = 213	*n* = 120	*n* = 110	-
Type of IGRA	QuantiFERON-TB Gold In-Tube	QuantiFERON-TB Gold In-Tube	LIOFeron TB/LTBI	-
IGRA and/or TST positivity, *n* (%)				
- PLHIV	59 (22.9%)	-	-	0.08 **
- HIV-uninfected individuals	35 (16.4%)	-	-	
IGRA positivity, *n* (%)				
- PLHIV	49 (19.1%)	15 (25.4%)	6 (8.1%)	0.03 *
- HIV-uninfected individuals	16 (8.8%)	15 (12.5%)	18 (12.8%)	0.44 *
TST positivity, *n* (%)				
- PLHIV	22/243 (9.1%)	-	-	0.30 **
- HIV-uninfected individuals	24/198 (12.1%)	-	-	
Kappa IGRA/TST (95% CI)				
- PLHIV	0.28 (0.12–0.43)	-	-	-
- HIV-uninfected individuals	0.15 (−0.03–0.33)	-	-	-
Positivity rate: IGRA vs. TST				
- PLHIV	19.1% vs. 9.1%	-	-	0.001 **
- HIV-uninfected individuals	8.8% vs. 12.1%	-	-	0.21 **

* Vilnius and Siauliai Centres vs. Klaipeda Centre. ** PLHIV vs. HIV-uninfected individuals. TB: tuberculosis; PLHIV: people living with human immunodeficiency virus; CD4: cluster differentiation-4; HIV: human immunodeficiency virus; IGRA: interferon gamma release assay; TST: tuberculin skin test; CI: confidence interval.

**Table 3 pathogens-12-00990-t003:** Agreement between QFT-GIT and TST (≥5 mm).

Results	PLHIV	HIV-Uninfected Individuals
Positive TST and positive QFT-GIT	12	5
Negative TST and negative QFT-GIT	189	161
Negative TST and positive QFT-GIT	32	13
Positive TST and negative QFT-GIT	10	19
Kappa (95% CI)	0.28 (0.12–0.43)	0.15 (−0.03–0.33)
Total	243 (11 cases not returned for TST evaluation)	198 (3 cases not returned for TST evaluation)

PLHIV: people living with human immunodeficiency virus; HIV: human immunodeficiency virus; QFT-GIT: Quantiferon-TB Gold in-Tube; TST: tuberculin skin test; Kappa: Cohen’s kappa coefficient; CI: confidence interval.

## Data Availability

The data presented in this study are available on request from the corresponding author.

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
