# Peer review of "Latent Tuberculosis Infection and Associated Risk Factors among People Living with HIV and HIV-Uninfected Individuals in Lithuania"

_pathogens, 2023, doi:10.3390/pathogens12080990_

Round 1

Reviewer 1 Report

It is a study on a cross-sectional study was conducted from August 2018 to May 2022 in three geographically distinct Infectious diseases departments/centres – Vilnius University Hospital Santaros Klinikos, Republican Siauliai County Hospital and Klaipeda University Hospital – providing the vast majority of primary, outpatient and inpatient care to PLHIV in Lithuania. This study compared LTBI prevalence in PLHIV and risk factors associated with LTBI in a country with a low incidence of HIV, intermediate prevalence of TB, and a high BCG vaccination coverage.

It is a work of importance in public health, demonstrating that the prevalence of LTBI defined by positive IGRA and/or TST among PLHIV was higher compared to HIV-uninfected population: 20.5% vs. 11.1%. This fact emphasizes the need of treatment of LTBI in this population.

Lines 64 and 292 show an error in the formatting of references: "Reference Source Not Found".

Please make corrections before submitting the final version.

Reviewer 2 Report

Attached 

·         English edits required throughout the manuscript. Few examples of improperly formed sentences:

·         Line 297-298: “Within our study population, PLHIV were less likely to have a positive TST than HIV-uninfected persons even significant association has not been found.”

·         Line 313-314: “In our study the discordant results of QFT-GIT and TST led to only a fair agreement between the two tests among PLHIV and a slight agreement in HIV-negative population.”

Round 2

Reviewer 2 Report

The authors have addressed the comments satisfactorily.

Quality has improved